# Anomalous deep-red luminescence of perylene black analogues with strong π-π interactions

Ningning Tang [1], Jiadong Zhou [1] ✉, Liangxuan Wang [2,3], Matthias Stolte[4], Guojing Xie[1], Xinbo Wen [1], Linlin Liu [1], Frank Würthner [4] ✉, Johannes Gierschner [2] ✉ & Zengqi Xie [1] ✉

Perylene bisimide (PBI) dyes are known as red, maroon and black pigments, whose colors depend on the close π−π stacking arrangement. However, contrary to the luminescent monomers, deep-red and black PBI pigments are commonly non- or only weakly fluorescent due to (multiple) quenching pathways. Here, we introduce N-alkoxybenzyl substituted PBIs that contain close π stacking arrangement (exhibiting $d_{\pi-\pi} \approx 3.5$ Å, and longitudinal and transversal displacements of 3.1 Å and 1.3 Å); however, they afford deep-red emitters with solid-state fluorescence quantum yields ($\Phi_F$) of up to 60%. Systematic photophysical and computational studies in solution and in the solid state reveal a sensitive interconversion of the PBI-centred locally excited state and a charge transfer state, which depends on the dihedral angle ($\theta$) between the benzyl and alkoxy groups. This effectively controls the emission process, and enables high $\Phi_F$ by circumventing the common quenching pathways commonly observed for perylene black analogues.

Luminescence of organic dyes and pigments has fascinated scientists since the mid-19th century but retains enduring appeals till now. Numerous organic color pigments are deeply rooted in the archeological and historic contexts, and experience a renaissance in organic electronics[1,2], such as organic light-emitting diodes (OLEDs)[3,4], dye lasers[5,6], solar concentrator[7] and biological imaging[8,9]. The rapid development of the organic electronics field has successfully explored abundant building block using common pigments such as perylene-3,4:9,10-bis(dicarboximides) (PBIs)[10–13].

PBI derivatives, which cover various shades in the solid state from red to black as commercial pigments, have favorable (opto-)electronic properties and high thermal and (photo-)chemical stability[2,14–16]. More than for most other pigment chromophores, the color of PBI solids depends much on the longitudinal and transverse displacement of neighboring dye molecules[17–20]. In particularly, pigments with significant longitudinal but small transverse intermolecular slip, as for instance observed for the commercial pigment Perylene Black 32 (PBk32), are able to absorb all the visible light. The broad band absorption, which is responsible for the black appearance, was ascribed to short range charge transfer (CT) interactions[21,22], which modulates the classical Kasha molecular exciton model, based on long-range Coulombic interactions, with its distinction in J- and H-aggregates[23,24]. In any case, other factors may play a role such as shadowing effects in samples of high optical density[25], and polarization effects[26], which can strongly modulate the absorption band shapes as a function of the angle of observation[27]. Despite the interesting absorption properties of black PBI pigments, however, these materials usually suffer from luminescence quenching in the solid state. The

[1]State Key Laboratory of Luminescent Materials and Devices, Institute of Polymer Optoelectronic Materials and Devices, Guangdong Provincial Key Laboratory of Luminescence from Molecular Aggregates, South China University of Technology, 510640 Guangzhou, P. R. China. [2]Madrid Institute for Advanced Studies, IMDEA Nanoscience, Ciudad Universitaria de Cantoblanco, C/ Faraday 9, 28049 Madrid, Spain. [3]Institute of Physical and Theoretical Chemistry, Eberhard Karls University Tübingen, 72076 Tübingen, Germany. [4]Institut für Organische Chemie and Center for Nanosystems Chemistry, Julius-Maximilians-Universität Würzburg, Am Hubland, 97074 Würzburg, Germany. ✉e-mail: zhoujd@scut.edu.cn; wuerthner@uni-wuerzburg.de; johannes.gierschner@imdea.org; msxiez@scut.edu.cn

quenching processes may include competitive pathways like singlet fission[28–32], symmetry-breaking charge separation[33–35], and formation of excimer traps[36], which are all closely related to the specific intermolecular arrangement in these materials. In fact, in a microscopic view, the mixed electronic states of two neighboring identical chromophores directs the exciton fate[37].

In general, avoiding strong π-π interaction and isolating the chromophore appeared as a suitable solution for high solid-state fluorescence[25,38–42]. Unfortunately, however, such chromophore isolation reduces the usefulness of these solid-state materials for those applications where charge carrier transport is required. Therefore, we focused our attention on stacked PBIs and indeed could demonstrate recently that a polymorph of PBk32, i.e. PBI substituted with 4-methoxybenzyl groups at the peripheral nitrogen atoms (**mb-PBI**), exhibits a close-to magic-angle stacking mode, and shows solid-state fluorescence with over 0.50 eV redshift relative to the monomer owing to the strong π-π interactions[43]. Most notably, the crystalline film exhibited appreciable near-infrared (NIR) emission with a fluorescence quantum yield $\Phi_F$ of about 10%. Therefore, the complex mb-PBI structure concept appeared very promising to generate emissive perylene black analogues, compared to earlier attempts with less complex structure modulation.

In the current work, we therefore systematically elucidate a series of PBI derivatives with different length of branched alkoxy chains at the *para*-position of the *N*-benzyl rings, also offering better solubility in comparison with **mb-PBI**. In solution, the compounds are all moderately emissive ($\Phi_F$ of 20% in toluene) to low emissive ($\Phi_F$ of 1–2% in dichloromethane (DCM)) due to the presence of a low-lying CT state. In crystalline films, deep-red (DR) emission with solid-state luminescence enhancement[44] is observed, instead of quenching by the predicted excitonic relaxation to multi-exciton states, charge-separated states or excimers in such π-stacked arrangement of PBI chromophores[45]; however, the enhancement depends largely on the length of the alkoxy substituents, as well as morphological factors, so that $\Phi_F$ in films ranges from 2% to up to 60%. This is considered as a specific asset of the compounds reported here, as bright DR/NIR solid-state emitters are still a major challenge in the field, also beyond PBIs[46,47]. Systematic photophysical studies in solution and solid state under varying environmental conditions, combined with computational studies fully rationalize the experimental results. Accordingly, we have discovered a long sought-after packing structure for this important class of pigments, now providing highly fluorescence solid state DR emitters.

## Results

### Synthesis

Derived from the parent compound **mb-PBI**, the methoxyl groups were replaced by branched alkoxy chains with different lengths, giving **4,6-ab-PBI**, **6,8-ab-PBI** and **8,10-ab-PBI** (Fig. 1a). All three alkoxybenzyl substituted PBIs were synthesized by a straightforward method with yields of 50% to 70% according to the previously reported procedures[15,48], and were characterized by ¹H NMR and ¹³C NMR (Supplementary Figs. 2–13 in the Supplementary Information; SI). The crude products were purified by column chromatography on silica gel using the mixture of petroleum ether and DCM as eluent. All compounds are well soluble in common organic solvents like toluene, chloroform and chlorobenzene but less soluble in polar solvent such as methanol. The detailed synthetic procedures and characterization data of the PBIs are given in the SI.

### Crystal structures

The molecular structures and stacking patterns of the PBIs were determined by single crystal X-ray diffraction measurements. Bright red single crystals of PBIs were successfully obtained by the liquid-liquid diffusion method, where ethanol (poor solvent) diffused slowly

into chloroform solution (good solvent) at room temperature for ~3 days. Plate-like crystals formed at the bottom of the solution (Fig. 1b). Suitable single crystals were selected, and measured in a nitrogen stream at 150 K. The crystal data and structure refinement are summarized in Supplementary Table 1.

All three PBIs showed similar geometries, consisting of a rigid and planar backbone (Fig. 2), and N-substituted benzyl groups pointing to the opposite sides, which resulted in a centrosymmetric chair conformation (Fig. 2c). The chiral carbon atoms of the alkoxy chains in the centrosymmetric positions led to an optical mesomer. A significant difference in the molecular structures is seen in the dihedral angle between the benzyl and alkoxy groups ($\theta$), giving 9° for **4,6-ab-PBI**, but 33° and 26° for **6,8-ab-PBI** and **8,10-ab-PBI** respectively, which indeed influenced the electronic transitions (*vide infra*).

All PBIs are aligned in a slipped π-stacked (face-to-face) fashion with a π-π distance of about 3.5 Å and large π-π overlap. In fact, about half of the π-surface overlaps with that of the neighbor molecules (Fig. 2b). The longitudinal ($X$), transversal ($Y$) slips are given in Table 1, giving counter pitch angles ($\delta$) of 49–50° for **4,6-ab-PBI**, **6,8-ab-PBI** and **8,10-ab-PBI**, respectively. Neighboring slipped-stacked columns are oriented in a skew angle of 94–97°, giving a zig-zag arrangement following the general classification scheme of ref. 25. Between the stacks, multiple intermolecular C-H⋯O interactions are found with distances of 2.5–2.8 Å, corresponding indeed to the expected O⋯H van-der-Waals distance of 2.7 Å, see Supplementary Fig. 14. Along the long molecular (N-N) axis direction, the hydrophobic interactions by branched alkoxy chains are expected to be much weaker compared to the π-π and H-bonding interactions, indicating that this direction was perpendicular to the slow growing faces, which therefore dominates the lamellar morphology. Therefore, the lamellar structure was readily assembled on the spin-coated film with post-annealing, which was verified by the orientation analysis form grazing-incidence wide-angle X-ray scattering (GIWAXS) measurements (Supplementary Fig. 22).

### Optical properties

The optical and photophysical properties of the three PBIs were measured both in dilute solution and in annealed spin-coated thin films with thickness of about 30 nm (Fig. 3). In solution, all three PBIs exhibit well-resolved absorption bands at 528, 490, and 460 nm, attributed to the apparent 0–0, 0–1 and 0–2 vibronic progression of the $S_0 \rightarrow S_1$ transition[49]; the latter corresponds to an excitation between the highest occupied and lowest unoccupied molecular orbitals (HOMO; LUMO) of the PBI core. Here, 0–0 is the most intense subband, as commonly observed in rigid π-core systems with little geometrical reorganization upon electronic excitation. Consequently, small Stokes shifts ($\Delta Ev_{Stokes}$) between absorption and fluorescence are observed; for instance, in chloroform 9 meV (**4,6-ab-PBI** and **6,8-ab-PBI**) and 13 meV (**8,10-ab-PBI**). Furthermore, the fluorescence spectra are approximately mirror-symmetrical to the absorption spectra.

Nevertheless, the fluorescence properties depend significantly on the environmental conditions. In non-polar solvents, such as toluene, the fluorescence spectra are similarly sharp as in absorption, and $\Phi_F$ are relatively high; i.e., 7–10% in chlorobenzene and 22–25% in toluene (Supplementary Figs. 15–17, and Supplementary Tables 4–6). In any case, the $\Phi_F$ are much smaller for the three compounds when compared with common PBIs, whose $\Phi_F$ are close to unity[14]. Concomitantly, the fluorescence lifetimes of the three compounds are significantly shortened against the common PBIs, giving about 1.6 ns in toluene. In contrast, in polar, aprotic solvents such as chloroform, DCM or tetrahydrofuran, the fluorescence spectra are significantly broadened vs. absorption, and hypsochromically shifted with respect to non-polar solvents. At the same time, $\Phi_F$ drops to only 1–2%, and the fluorescence decay in chloroform (Supplementary Fig. 18) is well-fitted by a bi-exponential decay with

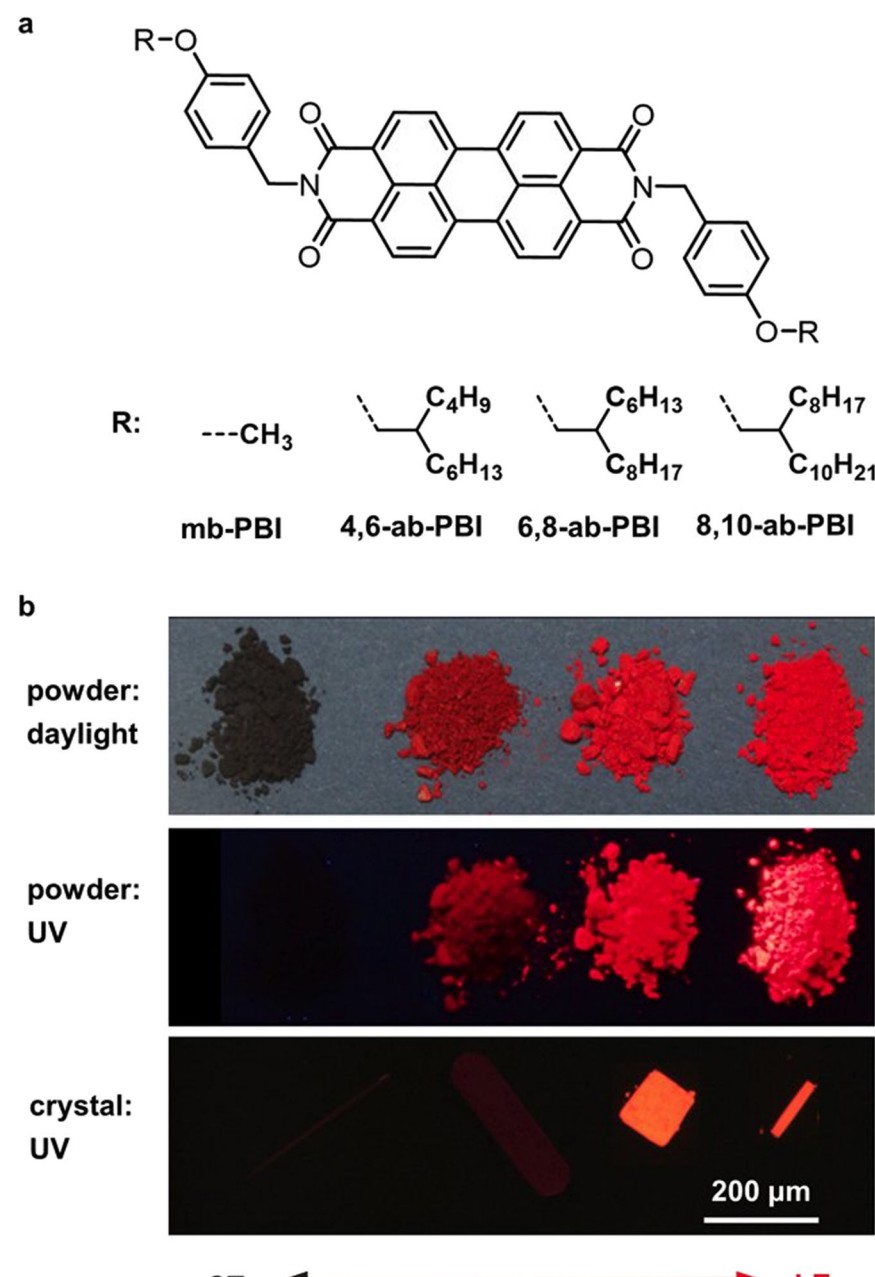

**Fig. 1 | Molecular structure and photographs of powder and crystals. a** Chemical structures of alkoxybenzyl substituted PBIs. **b** Photographs of the corresponding powders upon UV irradiation (365 nm) and daylight as well as fluorescence images of single crystals with exposure time of 158 milliseconds.

a short component of *ca*. 0.3–0.6 ns and a long one of 4–6 ns (Table 2).

The annealed spin-coated thin films show 'double-hump' absorption spectra with blue- and red-shifted components relative to the monomer spectra (Fig. 3). This common feature for black perylene pigments has been ascribed to the presence of low-lying CT states between closely stacked PBIs at longitudinal displacements of ~3 Å that afford two distinct bands due to strong Frenkel/CT-state mixing (Supplementary Table 3)[21,22], whose specific shape may further be affected by polarization effects (*vide supra*)[26,27]. The position of the high energy band is quite constant for the three compounds, located at *ca*. 470 nm, while the position of the low energy band largely differs, giving 628, 557/577 and 559 nm for **4,6-ab-PBI**, **6,8-ab-PBI**, and **8,10-ab-PBI**, respectively. This explains the somewhat darker red color of **4,6-ab-PBI** under daylight illumination compared to the other compounds (Fig. 1b). We emphasize in this context that **mb-PBI** has a black

appearance under daylight as the low energy band is significantly more red-shifted to *ca*. 680 nm in the film, so that the absorption onset is found at 800 nm[43], and all visible light is absorbed in the powder sample shown in Fig. 1b. Finally, the long extending tails of the absorption spectra in the red-part of the absorption spectra are entirely due to light scattering of the polycrystalline films, as verified by the comparison with the fluorescence excitation spectra in Fig. 3[14]. From this, the absorption edge can be estimated to about 700 nm, 650 nm and 660 nm for **4,6-**, **6,8-** and **8,10-ab-PBI**, respectively.

In contrast to the structured fluorescence spectra in solution, the fluorescence spectra of the films do not exhibit pronounced vibronic features, and reveal rather small widths of about 0.17 eV (FWHM), pointing to coherence effects as typically observed for J-type aggregates[14]. Due to the differences in the position of the low-energy absorption band, the bathochromic solid-state fluorescence shifts against solution are somewhat different, giving 0.59 eV, 0.48 eV and

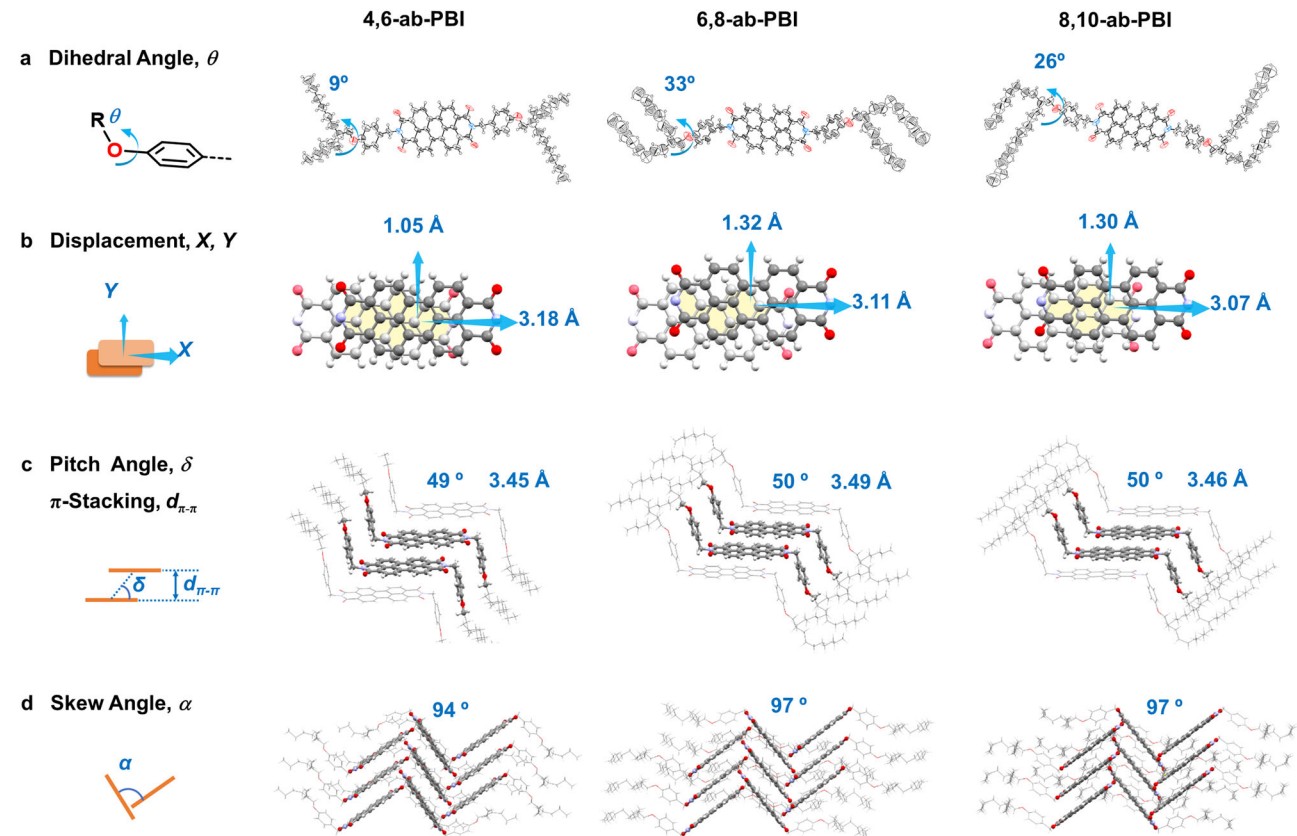

**Fig. 2 | X-ray crystal structures of the compounds under study. a** ORTEP drawings of the molecular structures (ellipsoid probability 50%), indicating the dihedral angle ($\theta$) between the benzyl and alkoxy groups. **b** Perspective view along the π-π stacking direction, indicating longitudinal ($X$) and transverse displacements ($Y$) between neighboring molecules. **c** Side view of the PBI π-core, indicating counter pitch angle ($\delta$) for the longitudinal slip and the distance between the π-planes ($d_{\pi\text{-}\pi}$). **d** View along the column axis, indicating the skew angle ($\alpha$) between neighboring columns.

0.53 eV for **4,6-ab-PBI**, **6,8-ab-PBI**, and **8,10-ab-PBI**, respectively. Because of the fluorescence emission in the DR, the emission color is very similar to the absorption color as in fact seen in the very similar hues of the powders under UV and daylight illumination (Fig. 1b). Significant differences are however observed for $\Phi_F$. As shown in Table 2, **4,6-ab-PBI** exhibits a very low fluorescence quantum yield in the film state of only 2% with a relatively short fluorescence lifetime of 0.7 ns. In sharp contrast, **6,8-ab-PBI** gives 27% and 1.9 ns, and $\Phi_F$ for **8,10-ab-PBI** reaches 60% and 6.1 ns for **8,10-ab-PBI**. This is surprisingly high as PBI derivatives with broad 'double hump' absorption features usually do not effectively emit in the solid state[28,30,50,51]. The pronounced difference between the low-emissive **4,6-ab-PBI** and the well-emissive **6,8-ab-PBI** and **8,10-ab-PBI** is also seen in the powder and crystal samples under UV irradiation (Fig. 1b)[25]; this excludes morphological reasons for the observed variance, but suggests differences in the intra- and/or intermolecular solid-state architecture.

## Discussion
### Solution properties
Different from many other common PBIs with $\Phi_F$ yield close to unity[14], the dyes studied here showed only weak emission in dilute solutions of

polar, aprotic solvents. In common PBIs, due to the fact that the nitrogen atoms exhibit orbital nodes for both HOMO and LUMO, substituents at the *N*-position hardly affect the characteristic optical $S_0 \rightarrow S_1$ transition; the latter is described as a locally excited (LE) state within the PBI core. In the current case however, the alkoxybenzyl substituents introduce two low-lying singlet excited CT states, formed between the frontier MOs of the alkoxybenzyl donor (D) and the PBI acceptor (A), see Fig. 4a and Supplementary Fig. 28. According to our time-dependent density functional theory (TD-DFT) calculations on the fully optimized structure, the CT states correspond to $S_{1,2}$ with vanishing oscillator strength $f_{1,2}$; the PBI-localized LE state is then found *ca.* 0.10 eV above ($S_3$; with $f_3 \approx 1.0$). After photoexcitation from the ground state (GS) to the LE state, photoinduced electron transfer (PeT) takes place, generating the formal CT state; this process is expected to be highly efficient at ambient temperatures due to the spatial proximity of D and A[52]. Due to the charge-separated nature of the CT state, subsequent back electron transfer recovers GS. Concomitantly, very small $\Phi_F$ are expected in this electronic situation in agreement with our experimental results in polar solution, and as indeed found in other PBI-based D-A dyads or D-A-D triads with short linkers[53,54].

From the simple PeT scenario, mono-exponential fluorescence decay is expected, where the PeT rate shows an exponential dependence on the D-A separation[52]. Interestingly however, in the current case, bi-exponential kinetics are observed (Table 2 and Supplementary Figs. 18–21). This suggests a somewhat more complex kinetic model, i.e., by repopulation of the LE state, which is thermally accessible from the CT state. In fact, the energy of the CT state depends sensitively on the rotation of the alkoxy group relative to the benzyl group ($\theta$) as the CT state is sufficiently stabilized only close to the co-planar conformation ($\theta = 0°$) due to the destabilization of the donor-localized occupied MOs

## Table 1 | Structural parameters extracted from the crystal structures, as illustrated in Fig. 2

|  | $X$ /Å | $Y$ /Å | $d_{\pi\text{-}\pi}$ /Å | $\delta$ / ° | $\theta$ / ° | $\alpha$ / ° |
|---|---|---|---|---|---|---|
| **4,6-ab-PBI** | 3.18 | 1.05 | 3.45 | 49 | 9 | 94 |
| **6,8-ab-PBI** | 3.11 | 1.32 | 3.49 | 50 | 33 | 97 |
| **8,10-ab-PBI** | 3.07 | 1.30 | 3.46 | 50 | 26 | 97 |

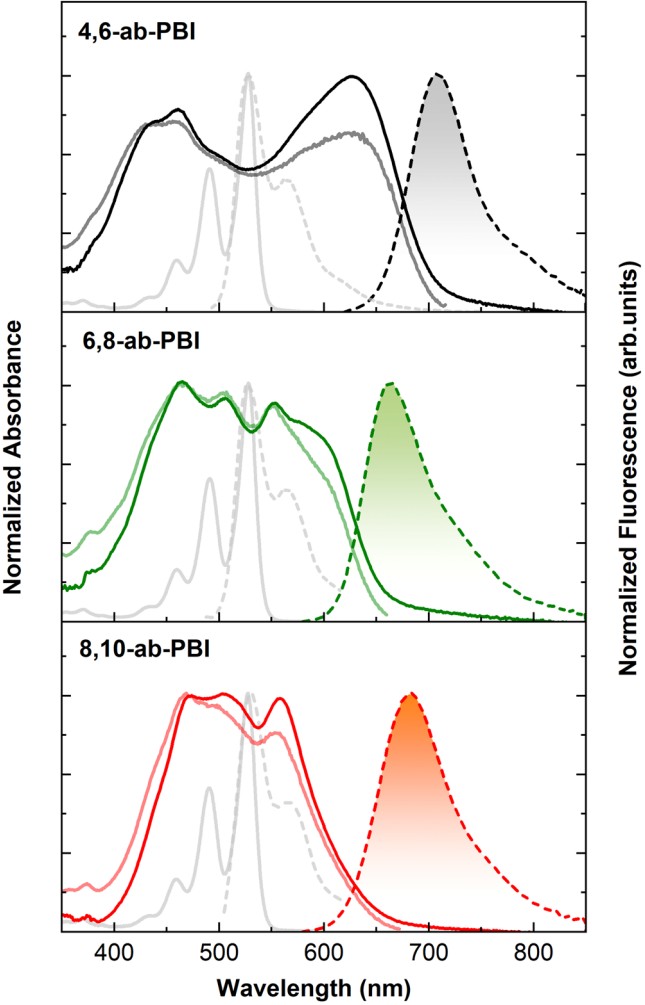

**Fig. 3 | Basic optical properties of 4,6-, 6,8-, 8,10-ab-PBI.** Normalized UV-vis absorption (solid lines), excitation (light solid lines) and fluorescence (dashed lines with filled area) spectra of **4,6-, 6,8-, 8,10-ab-PBI** (top to bottom) annealed spin-coated thin films on quartz substrates, respectively (Supplementary Fig. 22). Gray solid and dashed lines show the absorption and fluorescence spectra of the monomers in chloroform solutions (1 × 10⁻⁵ M for UV-vis absorption spectra and 1 × 10⁻⁷ M for fluorescence spectra).

**Table 2 | Optical and photophysical properties of the PBI derivatives in chloroform solution, and as annealed spin-coated PBI thin films**

| Sample | | $\lambda_{abs}{}^{a}$/nm | $\lambda_{em}{}^{a}$/nm | $\tau_F$/ns | $\Phi_F$/% |
|---|---|---|---|---|---|
| **4,6-ab-PBI** | Solution | 528 | 530 | 0.3 (40%), 4.1 (60%)[b] | 1 |
| | Film | 628 | 708 | 0.7 | 2 |
| **6,8-ab-PBI** | Solution | 528 | 530 | 0.3 (72%), 4.7 (28%)[b] | 1 |
| | Film | 557/577 | 668 | 1.9 | 27 |
| **8,10-ab-PBI** | Solution | 527 | 530 | 0.2 (52%), 4.3 (48%)[b] | 2 |
| | Film | 559 | 684 | 6.1 | 60 |

[a]The given values are those for the maximum of the largest wavelength absorption and emission band.
[b]Biexponential fits, obtained by deconvolution; fractional intensities are given in parentheses.

through the enhanced conjugation in the alkoxy-benzyl moiety; see Fig. 4b. Thus, at larger angles $\theta$, this can lead to a crossover of the CT and LE states (Fig. 4b). Due to the small separation $\Delta E_{LE\text{-}CT}$, this crossover can be thus induced by thermal fluctuations of $\theta$. At room temperature, the energy barrier of 0.01 eV corresponds to an equilibrium constant $K_{LE/CT} = 0.02$ which correspond well to the measured $\Phi_F$ values in Table 2 if we (reasonably) assume a non-emissive CT state and a fully emissive LE state. This 3-state model (i.e., GS, LE, CT) with communicating CT and LE states in fact agrees with the observed bi-exponential kinetics found in solution (Table 2), as such behavior is indeed expected for a consecutive-competitive reaction with a reversible step[55].

The energy separation $\Delta E_{LE\text{-}CT}$ depends critically on the solvent polarity; the latter commonly induces significant bathochromic shifts for CT states due to pronounced solvent reorganization in the excited state, which stabilizes the latter[56–58]. On the other hand, the LE state can experience small bathochromic[59], or hypsochromic shifts[58], depending on the change of ground and excited state dipole moments. Thus, $\Delta E_{LE\text{-}CT}$ should increase with solvent polarity, and should thus decrease $\Phi_F$; this indeed agrees with the experimental observations, which gave $\Phi_F \approx 1\%$ in DCM, but $\Phi_F \approx 23\%$ in toluene (Supplementary Figs. 18–21 and Supplementary Tables 4–7).

Upon cooling of the chloroform solution down to 190 K, the fluorescence intensity increases by about a factor of two, and elongates the mean lifetime, i.e., to about 1 ns; this is due to the increase of the (dominant) short lifetime, while the long lifetime changes little, however with a smaller statistical weight (Supplementary Figs. 19–21). We ascribe the increase of the lifetime to the commonly observed thermal activation of the PeT process, as described by the Marcus equation[52], to populate the CT state. The temperature dependence is however not very pronounced as the repopulation of the LE state is as well thermally activated. Furthermore, we note that also CT and LE state energies are expected to change somewhat upon cooling.

## Solid-state properties

In the solid state, the photophysics of π-conjugated molecules are often strongly modulated against solution by geometrical changes, by intermolecular interactions and by morphological factors, which all impact spectral positions and band shapes, as well as radiative and non-radiative rates[25]. For the PBIs under study, strong intermolecular interactions are evident from the pronounced bathochromic shift of the rather narrow fluorescence spectra vs. solution, originating from the PBI-centered LE state. Significant interactions are expected from the considerable π-π overlap as discussed further up. In PBI-type materials, depending on the specific region of the excited state potential hypersurface[21], π-stacked arrangements with such small transversal slips give usually rise to ultrafast nonradiative exciton deactivation via singlet fission, symmetry-breaking charge separation, or excimer formation[45]. Thus, the reported high fluorescence quantum yield and long lifetimes within the current PBI series, despite the similar structural arrangement, may surprise at the first glance.

Strong differences for $\Phi_F$ of the annealed films of the current PBI series are indeed observed, giving only 2% for **4,6-ab-PBI**, while **6,8-ab-PBI** and **8,10-ab-PBI** are highly emissive with 27% and 60%, respectively. Nevertheless, we note that the interpretation of solid-state fluorescence data is generally complicated by morphological issues. In fact, small domain sizes in polycrystalline samples may promote trapping[25]. Moreover, polymorphism is also frequently observed for π-conjugated organic materials, which may significantly impact $\Phi_F$[25]. Such morphological issues are also evident in the current case; in fact, $\Phi_F$ of powder samples vary from those in the annealed films, giving 2% for **4,6-ab-PBI**, while 17% and 15% are observed for **6,8-ab-PBI** and **8,10-ab-PBI**. In any case, the very pronounced difference between low-emissive **4,6-ab-PBI** and well-emissive **6,8-ab-PBI** and **8,10-ab-PBI** is evident also here, and furthermore in the single crystal samples under UV illumination (Fig. 1b); in these homogenous

**Fig. 4 | Deactivation model for the PBI derivatives in solution according to single point TD-DFT calculations. a** Frontier MO energy levels and topologies for torsional angle $\theta$ between the alkoxy and benzyl groups of 0° and 90°. **b** Potential energy surface (PES) of GS, CT, and LE states along $\theta$. The energy levels indicated in the diagram is in eV.

0.1–1 mm crystals, trapping at surfaces or interfaces can be indeed excluded because of low defect concentration, small surface/volume ratio, and large average distance of the exciton to the surface[25]. In all, this suggests intrinsic, i.e., intra- and/or intermolecular, reasons for the pronounced drop of $\Phi_F$ in **4,6-ab-PBI** vs. **6,8-ab-PBI** and **8,10-ab-PBI**.

Evidently, as the π-stacking arrangement of the compounds does not significantly differ from each other, the reason for the largely varying $\Phi_F$ is thus not ascribed to intermolecular factors, but rather to intramolecular ones, i.e., to the difference of the angle $\theta$ between the alkoxy group relative to the benzyl group. Indeed, as our solution study clearly showed, this strongly impacts the photophysics. As a matter of fact, in single crystals of **6,8-ab-PBI** and **8,10-ab-PBI**, $\theta$ is rather large with 33° and 26°, respectively (Fig. 2 and Table 1); in sharp contrast, $\theta$ is small for **4,6-ab-PBI** (9°). This gives a plausible explanation for the observations in the thin films, consistent with our electronic scheme in Fig. 4. In **6,8-ab-PBI** and **8,10-ab-PBI** $\theta$ is large enough to promote the CT state toward high enough energy to prohibit the fluorescence quenching of the LE state. In fact, according to our calculations, the CT state is found 0.05 eV below LE at 26° and 0.02 eV at 33°, respectively; on the other hand, for **4,6-ab-PBI** with $\theta = 9°$, CT is calculated to be 0.09 eV below LE, so that here $\Phi_F$ remains small. Concomitantly, in **mb-PBI**, where $\theta = 2°$, the fluorescence is even further quenched. In any case, although the main factor for the large differences in $\Phi_F$ is seen in the variation of $\theta$, morphology is evidently contributing as well in the polycrystalline thin film samples. This is in fact seen in the significant smaller $\Phi_F$ in the thin film sample of **6,8-ab-PBI** compared to **8,10-ab-PBI**, despite the somewhat smaller $\theta$ in the latter.

Evidently, as in solution, quenching via PeT in the solid state is a thermally activated process, so that cooling of the films to 80 K significantly enhances $\Phi_F$ in particular for **4,6-ab-PBI** (25%, see Fig. 5); for **6,8-ab-PBI** and **8,10-ab-PBI**, $\Phi_F$ of about 70% and 80%, respectively, are observed under this condition. The fluorescence enhancement upon cooling is accompanied by an elongation of the fluorescence lifetimes in the same order of magnitude (Supplementary Figs. 23–25 and Supplementary Table 8), indeed indicating the suppression of the nonradiative quenching pathway upon cooling. The observed bathochromic shift of the LE emission upon cooling (Fig. 5b) is ascribed to the increased effective polarizability[60,61]; this stabilization of the LE state may additionally contribute to the fluorescence enhancement upon cooling. In fact, there is a systematic trend on the enhancement factor $\Phi_F$ (80 K)/$\Phi_F$ (300 K) with the extent of the bathochromic shift,

being the largest for **4,6-ab-PBI** (Fig. 5b). Finally, it is remarked that the torsional angle is not expected to be constant with temperature. This is evident from ab initio molecular dynamics (AIMD) simulations (Supplementary Figs. 26–27 for details); these evidence negligible intermolecular dynamics, but significant variations of dihedral angles in the solid state. Importantly, no major differences in the dynamics are observed between the three different compounds, as the mean deviation from the respective equilibrium values of $\theta$ are rather smaller in **4,6-** compared to **6,8-** and **8,10-ab-PBI**, see (Supplementary Fig. 27). This excludes a dynamic origin for the difference between the low emissive **4,6-** vs. the highly emissive **6,8-** and **8,10-ab-PBIs**, i.e., by a less restricted conformational space in the former. Therefore, the AIMD results further support our scenario on a specific locking of distinct dihedral angles $\theta$ for the different compounds, which regulates the access of the emissive LE state.

To summarize, we have reported a series of *N*-alkoxybenzyl substituted perylene bisimides (**4,6-**, **6,8-**, **8,10-ab-PBI**) with different alkoxy substitution pattern, which showed a strong dependency of the fluorescence quantum yields $\Phi_F$ on the environmental conditions. While in polar, aprotic solvents all compounds were weakly emissive, they became considerably emissive upon cooling, as well as in non-polar solvents. In the solid state, $\Phi_F$ was low for **4,6-ab-PBI** and high for **6,8-** and **8,10-ab-PBI**. A significant increase of $\Phi_F$ was observed upon cooling for all compounds. All observations could be consistently explained by a model based on an emissive PBI-core-centered LE state vs. a low-lying non-emissive CT state formed between the PBI acceptor (A) and the alkoxybenzyl donor (D) with a distinct dihedral angle $\theta$ between the benzyl and alkoxy groups. The CT state is accessed from the LE state by fast thermally activated photo-induced intramolecular electron transfer (PeT), which can repopulate LE. The equilibrium between LE and CT is subtly influenced by temperature, polarity and the angle $\theta$, which allowed to rationalize the observation in solution under the various conditions. In the solid state, the anomalous high fluorescence efficiency and longer lifetime indicated the suppression of nonradiative exciton deactivation through multi-exciton states, charge-separated states or excimers, which are otherwise commonly observed in PBI chromophores with such pronounced π-stacked arrangement. On the other hand, varying $\Phi_F$ in the current PBI series is ascribed to intramolecular factors, where the much smaller dihedral angle $\theta$ for **4,6-ab-PBI** in comparison with **6,8-**, **8,10-ab-PBI** rationalizes the low $\Phi_F$ of the former vs. the high $\Phi_F$ of the latter. The study

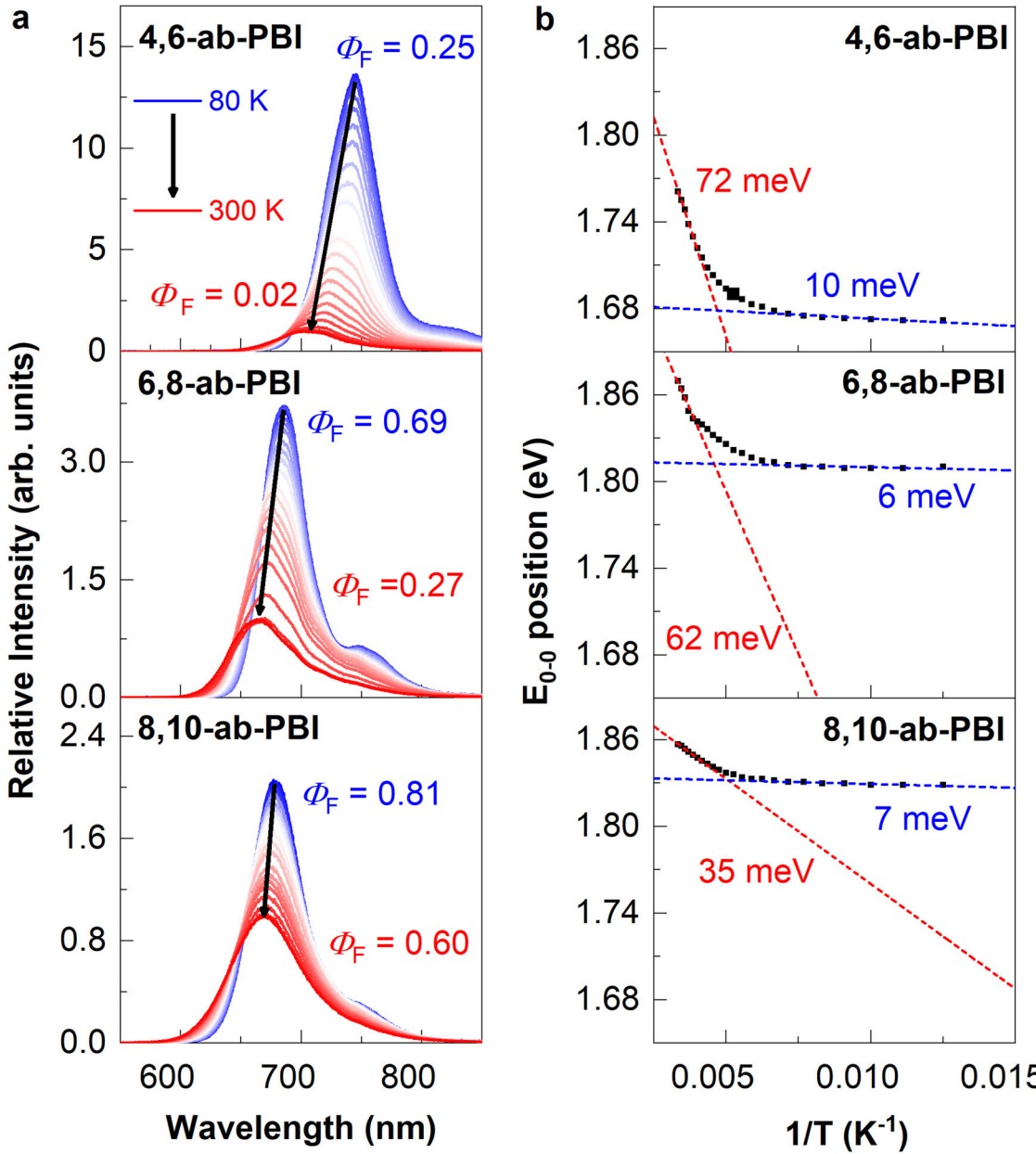

**Fig. 5 | Temperature-dependent steady-state fluorescence spectra of PBIs crystalline films. a** Temperature-dependent steady-state fluorescence spectra of PBIs crystalline films (heating from 80 K to 300 K, $\lambda_{ex}$ = 480 nm). **b** Temperature dependence of 0–0 emission peak energy (Supplementary Figs. 23a–25a) and the Gaussian energetic disorder calculated according to the slopes at each extreme along with the temperature-dependent fluorescence spectra.

showed that complex D-A-D PBI triads enable to modulate solution and solid-state fluorescence properties to a large extent. Most importantly, this allows to create highly desired emissive DR solid-state emitters, despite close π-stacking, which commonly leads to quenching in perylene black analogues. Future research will be dedicated to the exploration of how to systematically tune $\theta$ and thus $\Phi_F$ by targeted design of the molecular backbone and the (alkyl) substituents, as well as on the application of such highly emissive solid-state materials in organic electronic devices as for instance in light-emitting field effect transistors and solar concentrators.

## Methods
### Sample preparation
The synthesis and analysis of single-crystal X-ray diffraction of PBIs are described in the Supplementary Information. All raw materials were purchased from commercial sources and used without further purification, unless otherwise noted. Column chromatography was performed on silica gel (Hai Yang Silica 60, particle size 200–300 nm). NMR spectra were recorded on a Bruker-500 spectrometer at 500 MHz in deuterated solvents at 298 K. Chemical shifts were reported as $\delta$ values (ppm) relative to the residual solvent signal.

Films were prepared by spin-coating (2500 rpm, 30 s) from solution (chloroform for **4,6-ab-PBI** and **6,8-ab-PBI**, chlorobenzene for **8,10-ab-PBI**) onto substrates, and were further annealed in air (**4,6-ab-PBI** and **6,8-ab-PBI** at 180 °C for 30 min, **8,10-ab-PBI** at 160 °C for 30 min). Single crystals were successfully grown by liquid-liquid diffusion method of ethanol into chloroform solutions.

### X-ray diffraction
Suitable single crystals were selected and the X-ray diffraction data were collected in a Rigaku XtaLAB P2000 FR-X setup with a rotating copper anode and a Pilatus 200 K detector.

Grazing-incidence wide-angle X-ray scattering (GIWAXS) measurements were performed in a Xenocs Xeuss 2.0 system, equipped with a MetalJet-D2 Excillum source ($\lambda = 1.34144$ Å); the beam size was $0.8 \times 0.8$ mm$^2$ in high flux mode. The scattered signals were collected by a Pilatus3R 1 M detector, Dectris (Pixel Size: $0.172 \times 0.172$ mm$^2$). A movable beamstop was deposited at direct beam position to protect the detector from saturation within a long exposure time (1800 s).

### UV-vis absorption and fluorescence measurements

The UV-vis spectra of $1 \times 10^{-5}$ M solutions were carried out with a Shimadzu UV-3600 spectrometer. Steady-State fluorescence excitation and emission spectra of solutions with maximum extinction ~0.1 and of the spin-coated films (*ca.* 30 nm) were recorded in an Edinburgh FLS980 spectrometer. All spectra were corrected for the characteristics of the lamp source and detection unit, respectively.

Absolute fluorescence quantum yields were obtained in a calibrated integrating sphere (Hamamatsu, C11347-11). Fluorescence images of the single crystals were recorded using a Leica DM4000 microscope with an EL6000 light source and excitation filters.

Temperature-dependent steady-state spectra and time-resolved fluorescence spectra were recorded by homebuilt optical paths. Film and solution samples were placed inside a vacuum nitrogen cryostat (OptistatDN, Oxford). Heating of the film samples was carried out from 80 K to 300 K in vacuum, using a Mercury iTC temperature controller. The samples were excited with a femtosecond laser pulse at 480 nm with a pulse duration of 80 fs and a repetition rate of 80 MHz (Spectra Physics, Mai Tai & Inspire HF 100). Steady-state spectra were obtained by integration with a CCD camera (Princeton Instruments, Pixis 100B). Fluorescence time traces were recorded by time-correlated single photon counting, using a monochromator (Princeton Instruments, Acton 2300i) and a PicoQuant HydraHarp 400 system).

### DFT calculations

Geometry optimizations of the molecules were carried out at the DFT level of theory without geometry restriction, as well as with fixed dihedral angles $\theta$ between the benzyl and alkoxy groups. Single point TD-DFT calculations were performed to obtain the vertical transition energies in implicit solvent (DCM) through the polarizable continuum model (PCM). In all cases, the B3LYP functional with the standard 6−311 G* basis set with dispersion correction (D3) was applied, as defined in the Gaussian16 program package[62]. While standard functionals like B3LYP are known to fail for states with partial CT character[63], they operate well for states with very pronounced CT character (as in the current case)[64,65]; a comparison with long-range corrected functionals is given in the SI. It is noted that the computational modelling of solvent effects for CT states is non-trivial as the adiabatic approximation (on which the PCM model is based) does not necessarily holds, as discussed in ref. 58. It is further noted that the excited state ordering in the TD-DFT calculations at $\theta = 0°$ differs from the one-electron configurations in DFT due to configuration interaction. In fact, in the DFT-based MO diagram in Fig. 4 the donor-located occupied MO are lower than the acceptor-located ones; however, in the TD-DFT description, this situation reverses.

The geometry of the dimers was optimized by means of a QM/MM (quantum mechanics/molecular mechanics) approach using the ONIOM implementation in Gaussian16. Here, the torsion angles were fixed to those extracted from the experimental X-Ray data at 150 K and 300 K to give a reliable crystal environment. Electronic transitions were performed by single-point TD-DFT calculations.

The dynamics of the PBI derivatives in a realistic environment were carried out by a 4 ps AIMD simulations, using the ORCA 5.0 software[66-68] at the B3LYP-D3/6-31 G* level of theory at 150 K and 300 K, where the input files were prepared with the Multiwfn program[69,70], based on the initial geometries from the X-Ray data. The initial atomic velocity was set according to Boltzmann-Maxwell

distribution. The relative motion was investigated between nearest neighbor molecules, and quantitative statistics on the trajectory was performed.

## Data availability

All data generated in this study are provided in the paper and Supplementary Information, and the raw data supporting this study are available from the Source Data file. Source data are provided with this paper.

The X-ray crystallographic coordinates for structures reported in this study have been deposited at the Cambridge Crystallographic Data Centre (CCDC), under deposition numbers 2211699, 2211700, 1974107, 2211720, 2211721. These data can be obtained free of charge from The Cambridge Crystallographic Data Centre via www.ccdc.cam.ac.uk/data_request/cif. Source data are provided with this paper.

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

## Acknowledgements

We sincerely thank the financial supports from National Natural Science Foundation of China (NSFC) (21975076, 52003089, 52103206, 21733005, 51873068) (Z.X. and J.Z.), the Fund of the Key Laboratory of Luminescence from Molecular Aggregates of Guangdong Province (2019B030301003) (Z.X.). The work in Madrid was supported by the Spanish Ministerio de Economía y Competitividad (MINECO-FEDER project CTQ2017-87054), by the Severo Ochoa program for Centers of Excellence in R&D of the MINECO/MICINN (SEV-2016-0686, CEX2020-001039-S) and by the Campus of International Excellence (CEI) UAM + CSIC (J.G.). We also thank the support from the 111 Project (G2022163017L) (F.W. and Z.X.) and Science and Technology Projects in Guangzhou (202102020561, 202102020401) (J.Z.).

## Author contributions

Z.X. conceived the project and suggested the molecular structures. N.T. performed the chemical synthesis, grew single crystals, and carried out most of the characterization and data collection. J.Z. conducted X-ray diffraction measurements and GIWAXS measurement. L.W. and J.G. conducted the DFT calculations and analyzed the calculated results. G.X., X.W., and L.L. contributed to the construction of the laser path of temperature-dependent fluorescence spectra. M.S. confirmed the UV-vis spectra, PL spectra, lifetime and PLQY tests. N.T. and J.Z. wrote the original draft. F.W., J.G., and Z.X. supervised the work in the groups, and guided data interpretation, paper writing and revisions. All the authors discussed the results and commented on the paper.

## Competing interests

The authors declare no competing interests.
