## [Peer Review File · Nature Communications]

Anomalous Deep-Red Luminescence of Perylene Black Analogues with Strong π - π InteractionsReviewers' Comments:

Reviewer #1:

Remarks to the Author:

In this work, Tang et al. found interesting observations that N,N'-alkoxybenzyl-substituted perylene bisimide (PBI) crystals and polycrystalline thin films can show very high fluorescence quantum yields of about 60%, which is not common and also was barely achievable in other PBI families. This is because of various excited-state relaxation pathways from the initial photoexcited Frenkel state, such as excimer, symmetry-breaking charge separation (transfer), or singlet fission, which generally act as fluorescence quenching channels. As the authors pointed out as well, the realization about controlling chromophore packing arrangements at a molecular level is highly important since it plays a role in determining photophysical/chemical properties, which can be manipulated by comprehensive long-range Coulombic and short-range charge-transfer interactions. Among various types of excited-state relaxation pathways described above, here, the authors aimed to engineer PBI solids that show superior fluorescent properties for the application, like light-emitting field effect transistors. The reference compound in this study is a PBI molecule with 4-methoxybenzyl groups substituted at the imide positions, which has been known as perylene Pigment Black 32. Just by replacing methoxy groups with longer alkoxy ones, the authors could achieve high fluorescence quantum yields of crystals and films without changing PBI stacking structures a lot, and this was theoretically supported by potential energy curves along the alkoxy rotation coordinates. In my opinion, the present work is very interesting and could attract a broad readership, and all experiments and calculations pretty well support the authors' claim. Thus, I can recommend this manuscript for publication in Nature Communications. But before the final publication, two minor issues listed below should be addressed.

1) In Table 2, the percentages of shorter and longer time constants in the solution phase seem to be reversed when I compared them with the data in Supplementary Fig. 17. It looks like the main decay components in all solutions are the faster sub-ns time constants.

2) The authors used a B3LYP functional to construct potential energy curves of locally-excited (LE) and charge-transfer (CT) states. As far as I know, the B3LYP functional is not ideal for calculating the charge transfer state since it does not well describe the non-Coulombic part of exchange functionals in the calculation. In this regard, I would like to hear the authors' opinions regarding how they are sure about the reliability of their calculation results. If possible, it is recommended that the authors should check one of other hybrid functionals, such as CAM-B3LYP, wB97XD, or BHLYP, to compare the results obtained with B3LYP.

Reviewer #2:

Remarks to the Author:

This manuscript reported a photophysical study of a series of unusually emissive PBI-based materials. Three PBI dyes containing N-alkoxybenzyl substituents with different branched alkoxy chains were synthesized. Analysis of their crystal structures reveals the significant influence of the lengths of the alkoxy chains on the dihedral angles between the benzyl and alkoxy groups (θ). All PBI dyes display low quantum yields in solution. However, a significant enhancement in the quantum yields of up to 80% is observed in the solid state for PBI dyes that have long alkoxy chains. The authors attribute this phenomenon to the interconversion between the PBI-based LE states and the alkoxybenzyl/PBI-based CT states.

The major contribution of this work is the demonstration that it is possible to achieve high fluorescence emission from PDI dyes in their highly aggregated states. This is unusual for common PBI dyes, which typically show fluorescent quenching upon aggregation. As a result, this work could be of interest to research fields related to the design and development of aggregation-induced emission chromophores in general.

The current paper focuses mainly on unraveling the underlying processes that lead to this unusual

photophysical phenomenon using a solid combination of experimental and computational work. However, it is possible that the PBI dyes in this manuscript could be just some special cases. To further improve its significance and impact levels, this work should also emphasize the structure/property relationship perspective as well.

- For the systematic study, symmetric branched alkoxy chains (e.g. 6,6-aa-PBI or 8,8-aa-PBI) or linear alkoxy chains could also be used. It is unclear to me why the authors chose asymmetric chains. This discussion should be included in the manuscript.

- Longer branched alkoxy substituents seem to increase the dihedral angle θ . What are the driving forces that lead to changes in these molecular arrangements?

- It is not obvious how the dihedral angle θ affects the energy level of the CT state? Please elaborate further on this point

In my opinion, these discussions will be helpful for the general readership of Nature Communications. The current manuscript is more suitable for specialized journals.

As already discussed in the introduction, the strategies based on weakening π - π stacking or isolating the chromophores can provide high solid-state luminescence but will be unsuitable for charge transport applications. I think the authors should examine the electrical properties of these PBI dyes and whether high electron mobility could still be obtained. This experiment will highlight the importance of having strong π - π stacking in these materials, which will greatly strengthen the manuscript.

In conclusion, this manuscript should be thoroughly revised for publication. Below are additional questions:

- I wonder why the aggregates of 6,8-ab-PBI do not show the highest quantum yield, even though its dihedral angle θ is the largest among the three PBIs, which should maximize the separation between the LE and CT states. Or is there an optimal dihedral angle θ that will best minimize the fluorescent quenching?

- Does the dihedral angle θ affect the orientation of the alkoxybenzyl ring relative to the PBI core?

- There is a crossover of LE and CT states in Figure 4b. Should Figure 1a include this point as well?

- On page 22, please check again the operating frequency of the NMR. Now, it is 500 MHz on a 300 MHz spectrometer.

Reviewer #3:

Remarks to the Author:

This is a research article about three perylene bisimide (PBI) derivatives and their photophysical properties. The PBIs show very low PLQY in solution, due to the charge transfer (CT) state. However, their photophysical properties changed a lot in the solid-state, including the red-shift of their PL spectra, broadened absorption spectra, and increased PLQY.

Other than the interesting photophysical properties, the paper did not demonstrate any applications of the PBIs. The discussion of the experimental observation was very insufficient. Some discussions lack quantitative evidence. In addition, there are some issues with the data presented, and some experimental and computational results should have been given in the main text. Overall, I don't see the importance of the PBIs, the explanations are not satisfying, and the relation of the molecular structure to the optical properties is not clear. Therefore, I decide to reject the paper for publishing in Nature Communications.

Here are my comments:

1. I would suggest the paper demonstrate some real applications of the PBIs. The paper did not convince me that the PBIs are useful materials.

2. I would suggest the paper show the crystal structures of all three PBIs in the main text, particularly

their packing patterns. Like what is in Figure 2b c and d.

3. What are the energy of GS, LE and CT states when the angle theta is 9, 33 and 26 degrees respectively? The paper should give more quantitative analysis rather than the sentence '...while in 6,8-ab-PBI and 8,10-ab-PBI θ is large enough to promote the CT state towards high enough energy to prohibit the fluorescence quenching of the LE state,' in page 18. What is a 'large enough' theta, and what is 'high enough' energy?

4. Can we tune the theta angle (and the optical properties) by tuning the molecular structure of PBIs? Or it is just a random configuration?

5. In addition, theta of 8,10-ab-PBI (26 degree) is smaller than theta of 6,8-ab-PBI, why the PLQY of 8,10-ab-PBI is much higher in the solid-state?

6. The 'emissive deep-red solid-state emitter' is a selling point of the PBIs. How do they compare with other deep-red emitters in either crystal or in solid matrix?

7. Is it possible that the optical properties of PBIs in solution and solid-state are simply due to the typical AIE effect. The low PLQY in solution is due to the rotation of the phenyl and the side-chains. While in the solid-state, the PLQY raises due to the restriction of the rotation. Increasing the sizes of the side-chains will reduce the freedom of rotation and thus increase the PLQY. Same explanation can also fit to the observation in the low-temperature measurements,

8. In SI, Figure S18b, S19b, S20b, S22b, S23b, S24b, the IRF should be provided. The time scale is not long enough that many decay profiles are not back to the baseline before the next pulse, in particularly S22b, S23b, S24b. I would suggest to re-do the measurement with lower repetition rate and add the IRF.

9. Why did the paper not show any fitting curve of the decay profiles?

10. There is clearly some raising component in S22b, S23b, S24b, as they are not showing sharp peaks at the excitation pulse (if you show the IRF). What is the raising component in the lifetime profiles? Is it the repopulation of LE?

Reviewer # 1

General Comments:

In this work, Tang et al. found interesting observations that N,N'-alkoxybenzyl-substituted perylene bisimide (PBI) crystals and polycrystalline thin films can show very high fluorescence quantum yields of about 60%, which is not common and also was barely achievable in other PBI families. This is because of various excited-state relaxation pathways from the initial photoexcited Frenkel state, such as excimer, symmetry-breaking charge separation (transfer), or singlet fission, which generally act as fluorescence quenching channels. As the authors pointed out as well, the realization about controlling chromophore packing arrangements at a molecular level is highly important since it plays a role in determining photophysical/chemical properties, which can be manipulated by comprehensive long-range Coulombic and short-range charge-transfer interactions. Among various types of excited-state relaxation pathways described above, here, the authors aimed to engineer PBI solids that show superior fluorescent properties for the application, like light-emitting field effect transistors. The reference compound in this study is a PBI molecule with 4-methoxybenzyl groups substituted at the imide positions, which has been known as perylene Pigment Black 32. Just by replacing methoxy groups with longer alkoxy ones, the authors could achieve high fluorescence quantum yields of crystals and films without changing PBI stacking structures a lot, and this was theoretically supported by potential energy curves along the alkoxy rotation coordinates. In my opinion, the present work is very interesting and could attract a broad readership, and all experiments and calculations pretty well support the authors' claim. Thus, I can recommend this manuscript for publication in Nature Communications. But before the final publication, two minor issues listed below should be addressed.

We sincerely appreciate for the positive evaluation of our work.

Comment 1: In Table 2, the percentages of shorter and longer time constants in the solution phase seem to be reversed when I compared them with the data in Supplementary Fig. 17. It looks like the main decay components in all solutions are the faster sub-ns time constants.

Author Reply: We thank the reviewer for the attention on the data. In any case, a glance at the experimental curves in polar solvents may be misleading; in fact, the percentages (fractional intensities) in Table 2 are a result of a deconvolution of the experimental (time-correlated single photon counting - TCSPC) data using the IRF. Comparison of the experimental curves with the IRF reveals that the curves at short lifetimes are dominated by the IRF. For this reason, the calculated fractional intensities are in fact small. In order to assist the readers, we now add in the Table " Biexponential fits, obtained by deconvolution; fractional intensities are given in parentheses".

Furthermore, we note that the data in Table 2 were recorded using a monochromator (Princeton Instruments, Acton 2300i) and a PicoQuant HydraHarp 400 system). Due to the limitation of the instrument (Spectra Physics, Mai Tai & Inspire HF 100, with a repetition rate of 80 MHz), the time scale is not long enough that the longer time constants in the solution phase are not back to the baseline before the next pulse. Therefore, we also repeated the measurements of lifetime in different solvents using an Edinburgh Instruments FLS980 spectrometer. Subsequently, we have replaced Supplementary Fig. 17 with the new data. The fitting results are now found in Supplementary Table 4-6. The new data are qualitatively similar to the former ones, giving a short lifetime of 0.3-0.6 ns and the long lifetime of 4-6 ns in polar solvents. The improved values are now also given in Table 2.

Supplementary Fig. 17 | Fluorescence decay curves for lifetime determination of **4,6-ab-PBI**, **6,8-ab-PBI** and **8,10-ab-PBI** in solvents of varying polarity. IRFs are given as grey lines; representative fitting curves are given in solid.

Comment 2: The authors used a B3LYP functional to construct potential energy curves of locally-excited (LE) and charge-transfer (CT) states. As far as I know, the B3LYP functional is not ideal for calculating the charge transfer state since it does not well describe the non-Coulombic part of exchange functionals in the calculation. In this regard, I would like to hear the authors' opinions regarding how they are sure about the reliability of their calculation results. If possible, it is recommended that the authors should check one of other hybrid functionals, such

as CAM-B3LYP, wB97XD, or BHLYP, to compare the results obtained with B3LYP.

Author Reply: The correct TD-DFT treatment of systems with CT character is a matter of ongoing discussions; in fact, it was shown that for states with partial CT character, standard DFT functionals like B3LYP tend to fail, see e.g. A. Dreuw, M. Head-Gordon, Chem. Rev. 2005, 105, 4009. On the other side, for states with very pronounced CT character (i.e. where the HOMO is localized in D and the LUMO in A), these functionals operate in fact better than long-range corrected functionals, see e.g. (a) Y. Olivier, J. C. Sancho-García, L. Muccioli, G. D'Avino, D. Beljonne, J. Phys. Chem. Lett. 2018, 9, 6149; (b) B. Milián-Medina, J. Gierschner, Org. Electron. 2012, 13, 895; (c) V. K. Singh, C. Yu, S. Badgujar, Y. Kim, Y. Kwon, D. Kim, J. Lee, T. Akhter, G. Thangavel, L. S. Park, J. Lee, P. C. Nandajan, R. Wannemacher, B. Milián-Medina, L. Lürer, K. S. Kim, J. Gierschner, M. S. Kwon, Nat. Catal. 2018, 1, 794. As the current molecules fall in the latter category, we utilized here the B3LYP functional. This indeed gives a much better reproduction of the spectral features in comparison with e.g. wB97XD or CAM-B3LYP, which we in fact did, but missed to report in the SI. This was now done (see Supplementary Table 10-13), and an explanation on the use of B3LYP was added in the Method part as follows:

" While standard functionals like B3LYP are known to fail for states with partial CT character,⁶³ they operate well for states with very pronounced CT character (as in the current case);^{64,65} a comparison with long-range corrected functionals is given in the SI."

Supplementary Table 10 | Calculation results using the PCM model, CAM-B3LYP functional with the standard 6-311G* basis set.

ES Transitions	0°			PBI-4,6 (9°)			PBI-8,10 (26°)			PBI-6,8 (33°)			90°		
	E _{vert} (eV)	Osc. Str. f (a.u.)	MOs Transition (Contr. in %)	E _{vert} (eV)	Osc. Str. f (a.u.)	MOs Transition (Contr. in %)	E _{vert} (eV)	Osc. Str. f (a.u.)	MOs Transition (Contr. in %)	E _{vert} (eV)	Osc. Str. f (a.u.)	MOs Transition (Contr. in %)	E _{vert} (eV)	Osc. Str. f (a.u.)	MOs Transition (Contr. in %)
S ₀ →S ₁	3.27	1.4041	H→L 90%	2.68	1.0855	H→L 98%	2.67	1.0890	H→L 98%	2.68	1.0871	H→L 98%	2.68	1.0877	H→L 98%
S ₀ →S ₂	4.57	0.0000	H-5→L 66% H→L+3 15% H-6→L+1 7%	3.78	0.0000	H-1→L 87%	3.81	0.0000	H-5→L 50% H-1→L 23% H-8→L 9% H→L+3 7%	3.81	0.0000	H-5→L 70% H→L+3 11% H-8→L 9%	3.81	0.0000	H-5→L 73% H→L+3 12% H-8→L 8%
S ₀ →S ₃	4.69	0.0000	H→L+2 44% H-8→L 41%	3.79	0.0000	H-2→L 91%	3.82	0.0000	H-1→L 63% H-5→L 23%	3.85	0.0000	H-1→L 82%	4.00	0.0000	H-8→L 48% H→L+2 28% H-1→L 8% H→L+3 6%
S ₀ →S ₄	4.80	0.2492	H-6→L 68% H→L+4 13% H-5→L+1 9%	3.82	0.0000	H→L+3 12% H-8→L 7%	3.83	0.0000	H-2→L 91%	3.87	0.0001	H-2→L 90%	4.01	0.1051	H-6→L 78% H→L+4 10%

Supplementary Table 11 | Calculation results using the PCM model, wB97XD functional with the standard 6-311G* basis set.

ES Transitions	0°			PBI-4,6 (9°)			PBI-8,10 (26°)			PBI-6,8 (33°)			90°		
	E _{vert} (eV)	Osc. Str. f (a.u.)	MOs Transition (Contr. in %)	E _{vert} (eV)	Osc. Str. f (a.u.)	MOs Transition (Contr. in %)	E _{vert} (eV)	Osc. Str. f (a.u.)	MOs Transition (Contr. in %)	E _{vert} (eV)	Osc. Str. f (a.u.)	MOs Transition (Contr. in %)	E _{vert} (eV)	Osc. Str. f (a.u.)	MOs Transition (Contr. in %)
S ₀ →S ₁	2.72	1.0888	H→L 97%	2.72	1.0896	H→L 97%	2.71	1.0932	H→L 97%	2.72	1.0911	H→L 97%	2.71	1.0920	H→L 100%
S ₀ →S ₂	3.85	0.0000	H-5→L 69% H→L+3 12% H-7→L 10%	3.85	0.0000	H-5→L 69% H→L+3 12% H-7→L 10%	3.85	0.0000	H-5→L 70% H→L+3 12% H-7→L 9%	3.85	0.0000	H-5→L 69% H→L+3 12% H-7→L 10%	3.84	0.0000	H-5→L 70% H→L+3 12% H-7→L 9%
S ₀ →S ₃	4.01	0.0000	H-7→L 37% H-1→L 22% H→L+2 21%	4.01	0.0000	H-7→L 38% H→L+2 22% H-1→L 21%	4.02	0.0000	H-7→L 39% H→L+2 25% H-1→L 13%	4.02	0.0000	H-7→L 43% H→L+2 26% H-1→L 11% H-12→L 6%	4.03	0.1093	H-6→L 76% H→L+4 11% H-5→L+1 7%
S ₀ →S ₄	4.02	0.0976	H-6→L 72% H→L+4 10% H-5→L+1 6%	4.03	0.0983	H-6→L 72% H→L+4 10% H-5→L+1 6%	4.03	0.1038	H-6→L 74% H→L+4 10% H-5→L+1 7%	4.03	0.1042	H-6→L 74% H→L+4 11% H-5→L+1 7%	4.04	0.0000	H-7→L 48% H→L+2 30% H→L+3 6%

Supplementary Table 12 | Calculation results using the PCM model, PBE0 functional with the standard 6-311G* basis set.

ES Transitions	0°			PBI-4,6 (9°)			PBI-8,10 (26°)			PBI-6,8 (33°)			90°		
	E _{vert} (eV)	Osc. Str. f (a.u.)	MOs Transition (Contr. in %)	E _{vert} (eV)	Osc. Str. f (a.u.)	MOs Transition (Contr. in %)	E _{vert} (eV)	Osc. Str. f (a.u.)	MOs Transition (Contr. in %)	E _{vert} (eV)	Osc. Str. f (a.u.)	MOs Transition (Contr. in %)	E _{vert} (eV)	Osc. Str. f (a.u.)	MOs Transition (Contr. in %)
S ₀ →S ₁	2.37	0.9631	H→L 100%	2.37	0.9650	H→L 100%	2.37	0.9619	H→L 100%	2.37	0.9684	H→L 100%	2.37	0.9687	H→L 100%
S ₀ →S ₂	2.46	0.0000	H-1→L 99%	2.46	0.0000	H-1→L 99%	2.50	0.0000	H-1→L 99%	2.54	0.0000	H-1→L 99%	2.97	0.0000	H-1→L 98%
S ₀ →S ₃	2.46	0.0025	H-2→L 99%	2.46	0.0016	H-2→L 99%	2.50	0.0085	H-2→L 99%	2.54	0.0003	H-2→L 99%	2.98	0.0003	H-2→L 99%

Supplementary Table 13 | Calculation results using the PCM model, BHHLYP functional with the standard 6-311G* basis set.

ES Transitions	0°			PBI-4,6 (9°)			PBI-8,10 (26°)			PBI-6,8 (33°)			90°		
	E _{vert} (eV)	Osc. Str. f (a.u.)	MOs Transition (Contr. in %)	E _{vert} (eV)	Osc. Str. f (a.u.)	MOs Transition (Contr. in %)	E _{vert} (eV)	Osc. Str. f (a.u.)	MOs Transition (Contr. in %)	E _{vert} (eV)	Osc. Str. f (a.u.)	MOs Transition (Contr. in %)	E _{vert} (eV)	Osc. Str. f (a.u.)	MOs Transition (Contr. in %)
S ₀ →S ₁	2.69	1.0916	H→L 98%	2.69	1.0928	H→L 98%	2.69	1.0947	H→L 98%	2.69	1.0947	H→L 98%	2.69	1.0956	H→L 98%
S ₀ →S ₂	3.71	0.0000	H-1→L 95%	3.71	0.0000	H-1→L 95%	3.80	0.0000	H-1→L 95%	3.80	0.0000	H-1→L 94%	3.89	0.0000	H-5→L 78% H→L+3 10% H-8→L 7%
S ₀ →S ₃	3.72	0.0002	H-2→L 96%	3.72	0.0002	H-2→L 96%	3.81	0.0001	H-2→L 96%	3.81	0.0001	H-2→L 96%	4.06	0.0000	H-8→L 36% H-1→L 30% H→L+2 24%

Reviewer # 2

General Comments:

This manuscript reported a photophysical study of a series of unusually emissive PBI-based materials. Three PBI dyes containing N-alkoxybenzyl substituents with different branched alkoxy chains were synthesized. Analysis of their crystal structures reveals the significant influence of the lengths of the alkoxy chains on the dihedral angles between the benzyl and alkoxy groups (θ). All PBI dyes display low quantum yields in solution. However, a significant enhancement in the quantum yields of up to 80% is observed in the solid state for PBI dyes that have long alkoxy chains. The authors attribute this phenomenon to the interconversion between the PBI-based LE states and the alkoxybenzyl/PBI-based CT states.

The major contribution of this work is the demonstration that it is possible to achieve high fluorescence emission from PDI dyes in their highly aggregated states. This is unusual for common PBI dyes, which typically show fluorescent quenching upon aggregation. As a result, this work could be of interest to research fields related to the design and development of aggregation-induced emission chromophores in general.

The current paper focuses mainly on unraveling the underlying processes that lead to this unusual photophysical phenomenon using a solid combination of experimental and computational work.

In my opinion, these discussions will be helpful for the general readership of Nature Communications. The current manuscript is more suitable for specialized journals.

Author Reply: We sincerely appreciate all comments about our work. Concerning the reports on AIE-type materials, it should be noted that the majority of these reports are dedicated to materials which avoid π -stacking. Here we show the unusual case of π -stacked chromophores, which do not show quenching in the polycrystalline state, regulated by side-chain engineering. We are convinced that our work will motivate a lot of groups to search for more examples of that kind.

Comment 1: However, it is possible that the PBI dyes in this manuscript could be just some special cases. To further improve its significance and impact levels, this work should also emphasize the structure/property relationship perspective as well. For the systematic study, symmetric branched alkoxy chains (e.g. 6,6-aa-PBI or 8,8-aa-PBI) or linear alkoxy chains could also be used. It is unclear to me why the authors chose asymmetric chains.

Author Reply: The choice of these side chain was motivated in the first place by our interest to increase the solubility of the PBIs (parent compounds known as insoluble pigments) and thereby to enable the casting of ordered thin films from solution. Further, we wanted the packing arrangement to remain in a narrow range, i.e. by keeping similar π - π -stacking structures of the PBIs. Indeed, the variation of the presented asymmetric chains generated three different compounds with the desired systematic variation of the packing structure with simple chain length variation, which directly related to their

luminous efficiency by adjusting the geometry of the donor/acceptor moieties. Future studies on unbranched and symmetric branched chains may explore to which degree this geometrical parameter can be tuned through specific packing effects.

Comment 2: Longer branched alkoxy substituents seem to increase the dihedral angle θ . What are the driving forces that lead to changes in these molecular arrangements?

Author Reply: It is evident, that the specific value for θ is a result of the molecular arrangement as imposed by geometrical demands (as well as weak van-der-Waals forces) of the alkyl groups. We don't know the exact interplay of driving forces, as this is a complex, delicate issue; however, we found some distinction when we have a close-up view onto the branched alkoxy chains in single crystal structures. In fact, in **6,8-ab-PBI** and **8,10-ab-PBI**, there are some additional non-covalent interactions between hydrogens at the end of the carbon chains. Besides, additional intermolecular C...H...O interactions were also observed between carbonyl and hydrogens that next to the oxygen on the alkoxy groups, which disappear in **4,6-ab-PBI** and may contribute to the larger dihedral angle θ in longer branched alkoxy substituents.

Comment 3: It is not obvious how the dihedral angle θ affects the energy level of the CT state? Please elaborate further on this point.

Author Reply: We thank the reviewer to stress this point. In order to help the readers in this point, we now extended the discussion of Fig. 4, writing:

"In fact, the energy of the CT state depends sensitively on the rotation of the alkoxy group relative to the benzyl group (θ) as the CT state is sufficiently stabilized only close to the co-planar conformation ($\theta = 0^\circ$) due to the destabilization of the donor-localized occupied MOs through the enhanced conjugation in the alkoxy-benzyl moiety; see Fig 4b."

Comment 4: As already discussed in the introduction, the strategies based on weakening π - π stacking or isolating the chromophores can provide high solid-state luminescence but will be unsuitable for charge transport applications. I think the authors should examine the electrical properties of these PBI dyes and whether high electron mobility could still be obtained. This experiment will highlight the importance of having strong π - π stacking in these materials, which will greatly strengthen the manuscript.

Author Reply: We agree on the importance of electric properties for this series of compounds, however faced challenges on our trials to obtain thin as well as elongated microcrystals for single crystal field-effect transistors as well as homogeneous layers for solution-processed OTFTs; therefore, this part of the work will be continued in our lab, and may be reported at a later stage.

Comment 5: I wonder why the aggregates of 6,8-ab-PBI do not show the highest quantum yield, even though its dihedral angle θ is the largest among the three PBIs, which should maximize the separation between the LE and CT states. Or is there an optimal dihedral angle θ that will best minimize the fluorescent quenching?

Author Reply: We agree with the reviewer, that this point has to be better explained. In fact, according to our calculations, we indeed expect an increased weight of the LE state with a continuous increase of θ ; we note though, that the angles for 6,8- and 8,10-ab-PBI are not so different. Thus, from these data, one may indeed expect rather similar Φ_F . The significant difference in the Φ_F for 6,8- and 8,10-ab-PBI, as noted by the reviewer, must be therefore ascribed to morphological factors which are surely present in the (polycrystalline) thin film samples. In fact, fluorescent microscope and SEM images show that 6,8-ab-PBI forms larger crystalline domains in the annealed thin films. This agrees with the common observation that longer branched alkyl chains are usually unfavorable for crystal growth.

In order to help the readers with these issues, this was now added in the main text, stating:

"In any case, although the main factor for the large differences in Φ_F is seen in the variation of θ , morphology is evidently contributing as well in the polycrystalline thin film samples. This is in fact seen in the significant smaller Φ_F in the thin film sample of **6,8-ab-PBI** compared to **8,10-ab-PBI**, despite the somewhat smaller θ in the latter".

Comment 6: Does the dihedral angle θ affect the orientation of the alkoxybenzyl ring relative to the PBI core?

Author Reply: The orientation of the benzyl ring with respect to PBI is 101° for **4,6-**, and 111° for **6,8-** and **8,10-ab-PBI**, respectively. This has however no notable impact on the state energies, as the D- and A-type MOs are largely separated, with D localized on the benzyl-alkoxy moiety, and A in the PBI chromophore core, respectively.

Comment 7: There is a crossover of LE and CT states in Figure 4b. Should Figure 1a include this point as well?

Author Reply: We thank the reviewer for the suggestion. To assist the readers in this, we have made some minor changes to Fig. 1, where a double arrow below Fig. 1b is used to illustrate a crossover of LE and CT states in Figure 4b.

Comment 8: On page 22, please check again the operating frequency of the NMR. Now, it is 500 MHz on a 300 MHz spectrometer.

Author Reply: We thank the reviewer for the attention on this; this was in fact a mistake and we now revised in the manuscript, reading

“NMR spectra were recorded on a Bruker-500 spectrometer at 500 MHz in deuterated solvents at 298 K.”

Reviewer # 3

General Comments:

This is a research article about three perylene bisimide (PBI) derivatives and their photophysical properties. The PBIs show very low PLQY in solution, due to the charge transfer (CT) state. However, their photophysical properties changed a lot in the solid-

state, including the red-shift of their PL spectra, broadened absorption spectra, and increased PLQY.

Other than the interesting photophysical properties, the paper did not demonstrate any applications of the PBIs. The discussion of the experimental observation was very insufficient. Some discussions lack quantitative evidence. In addition, there are some issues with the data presented, and some experimental and computational results should have been given in the main text. Overall, I don't see the importance of the PBIs, the explanations are not satisfying, and the relation of the molecular structure to the optical properties is not clear. Therefore, I decide to reject the paper for publishing in Nature Communications.

Author Reply: We sincerely appreciate all comments about our work. It is in fact correct that the current paper focused on the photophysical mechanism as the driving force for our unprecedented observation. We stress that such kind of fundamental insights are well in line with the scope of Nature Commun., which usually does not ask for immediate applications of novel observations (*vide infra*). Furthermore, we have given sufficient, convincing evidence for our model to explain the novel observation and cannot agree here with the reviewer's statement. In any case the reviewers' comments and suggestions allowed us to significantly improve our work. The reviewer should thus further consider our detailed answers to all reviewers, and the subsequent adds to the discussion in the manuscript, which should allow the readers to understand the underlying photophysics mechanism.

Comment 1: I would suggest the paper demonstrate some real applications of the PBIs. The paper did not convince me that the PBIs are useful materials.

Author Reply: It was shown in numerous earlier papers that PBI derivatives exhibit favorable (opto-)electronic properties and high thermal and (photo-)chemical stability, and could be applied in photoconductive systems, organic field-effect transistors, solar cells. However, PBI-based deep-red/near-infrared solid-state emitters with tightly π -stacked chromophores have not been disclosed till now and indeed considered impossible by many researchers due to the quenching effects of singlet fission, symmetry-breaking charge separation, and formation of excimer traps in the π -stack PBI system. Hence, the current work shows such emitters for strong π -stacking PBI moieties, and focus on the elucidation of the underlying photophysics mechanism of luminescence.

Indeed, there are many papers in top journals that exclusively elucidated photophysical properties of π -stacked PBI aggregates in thin films or in solution, such as a) Sung, J., Kim, P., Fimmel, B. et al. Direct observation of ultrafast coherent exciton dynamics in helical π -stacks of self-assembled perylene bisimides. *Nat Commun* **6**, 8646 (2015). b) A. K. Le, J. A. Bender, et al. Singlet Fission Involves an Interplay between Energetic Driving Force and Electronic Coupling in Perylenediimide Films, *J. Am. Chem. Soc.*

140, 814–826 (2018). c) Kim, T.W., Jun, S., Ha, Y. et al. Ultrafast charge transfer coupled with lattice phonons in two-dimensional covalent organic frameworks. *Nat Commun* **10**, 1873 (2019). d) Hong, Y., Kim, J. et al. Efficient multiexciton state generation in charge-transfer-coupled perylene bisimide dimers via structural control. *J. Am. Chem. Soc.* **142**, 7845–7857 (2020). e) Sebastian, E., Hariharan, M, Null exciton-coupled chromophoric dimer exhibits symmetry-breaking charge separation. *J. Am. Chem. Soc.* **143**, 34, 13769–13781 (2021). f) Lin, C., Kim, T., Schultz, J.D. et al. Accelerating symmetry-breaking charge separation in a perylenediimide trimer through a vibronically coherent dimer intermediate. *Nat. Chem.* **14**, 786–793 (2022). g) Hong, Y., Rudolf, M., Kim, M. et al. Steering the multiexciton generation in slip-stacked perylene dye array via exciton coupling. *Nat Commun* **13**, 4488 (2022), and so on.

Comment 2: I would suggest the paper show the crystal structures of all three PBIs in the main text, particularly their packing patterns. Like what is in Figure 2b c and d.

Author Reply: We thank the reviewer for the suggestion. We have put the detail molecular arrangement of all PBIs under study in Fig. 2 as follows:

Fig. 2 | X-ray crystal structures of 4,6- (a, b, c, d), 6,8- (e, f, g, h) and 8,10-ab-PBI (i, j, k, l). a, e, i ORTEP drawing of the molecular structures within the single crystal and the dihedral angle (θ) between the benzyl and alkoxy groups are included. The ellipsoid probability is 50%. **b, f, j** Perspective view along the π - π stacking direction. The longitudinal (X) and transverse displacements (Y) of neighboring molecules are indicated. **c, d, g, h, k, l** Perspective view from the side of the PBI π -core structure. The counter pitch angle (δ) for the longitudinal slip of adjacent molecules and the distance between the π -planes ($d_{\pi-\pi}$) as well as the skew angle (α) between neighboring molecular columns are marked.

Comment 3: What are the energy of GS, LE and CT states when the angle theta is 9, 33 and 26 degrees respectively? The paper should give more quantitative analysis rather than the sentence ‘...while in 6,8-ab-PBI and 8,10-ab-PBI θ is large enough to promote the CT state towards high enough energy to prohibit the fluorescence quenching of the LE state, ...’ in page 18. What is a ‘large enough’ theta, and what is ‘high enough’ energy?

Author Reply: Following the reviewer's recommendation, we now give the specific values in the text Page 20 and 21, reading

" In fact, according to our calculations, the CT state is found 0.05 eV below LE at 26° and 0.02 eV at 33°, respectively; on the other hand, for **4,6-ab-PBI** with $\theta = 9^\circ$, CT is calculated to be 0.09 eV below LE, so that here Φ_F remains small. "

Comment 4: Can we tune the theta angle (and the optical properties) by tuning the molecular structure of PBIs? Or it is just a random configuration?

Author Reply: The reviewer raised an interesting and relevant point. It is evident that the specific value for θ in the solid materials originates from the interplay of specific non-covalent interactions including π - π -stacking of the PBIs and van-der-Waals forces of the alkyl groups, but also additional other interactions (see discussion in the main text). The specific arrangement is however difficult to predict due to the typically flat energy landscape for these intermolecular interactions. In any case, we are planning to look into this issue in more detail in our future work. This is now remarked in the summary as follows.

" Further research will be dedicated to the exploration of how to systematically tune θ and thus Φ_F by targeted design of the alkyl substituents, ..."

Comment 5: In addition, theta of 8,10-ab-PBI (26 degree) is smaller than theta of 6,8-ab-PBI, why the PLQY of 8,10-ab-PBI is much higher in the solid-state?

Author Reply: This is indeed an important point. Please see the response to Comment 5 of reviewer 2 along with the additions to our revised manuscript.

Comment 6: The ‘emissive deep-red solid-state emitter’ is a selling point of the PBIs. How do they compare with other deep-red emitters in either crystal or in solid matrix?

Author Reply: We agree with the reviewer on the relevance of bright deep red emitters, also beyond PBIs. To stress this point, we now add in the end of the Introduction, after the statement on the high quantum yield:

" This is considered as a specific asset of the compounds reported here, as bright DR/NIR solid state emitters are still a major challenge in the field, also beyond PBIs.^{46,47} "

Comment 7: Is it possible that the optical properties of PBIs in solution and solid-state are simply due to the typical AIE effect. The low PLQY in solution is due to the rotation of the phenyl and the side-chains. While in the solid-state, the PLQY raises due to the restriction of the rotation. Increasing the sizes of the side-chains will reduce the freedom of rotation and thus increase the PLQY. Same explanation can also fit to the observation in the low-temperature measurements.

Author Reply: We agree with the reviewer on the observation of an 'AIE'-like effect, which we prefer to call in a more rigorous manner 'Solid State Luminescence Enhancement' (SLE), as already noted in the manuscript. However, we remind that the effect reported here is not trivial, and in particular cannot be explained in terms of a simple restriction of rotations: as discussed in the manuscript, it's the specific locking of the angle θ , which makes two compounds (**6,8-** and **8,10-ab-PBI** good solid-state emitters, and **4,6-ab-PBI** a bad one. Although there are in fact dynamic contributions for θ as demonstrated by our AIMD simulations, these calculations do not give any evidence for more dynamical freedom of **4,6-ab-PBI**, but rather less, see Fig. 25c in the SI. In order to assist the readers in this issue, after the part on AIMD, we now continue as follows

"Importantly, no major differences in the dynamics are observed between the three different compounds, as the mean deviation from the respective equilibrium values of θ are rather smaller in 4,6- compared to 6,8- and 8,10-ab-PBI, see (Supplementary Fig. 26). This excludes a dynamic origin for the difference between the low emissive 4,6- vs. the highly emissive 6,8- and 8,10-ab-PBIs, i.e. by a less restricted conformational space in the former. Therefore, the AIMD results further support our scenario on a specific locking of distinct dihedral angles θ for the different compounds, which regulates the access of the emissive LE state."

Comment 8: In SI, Figure S18b, S19b, S20b, S22b, S23b, S24b, the IRF should be provided. The time scale is not long enough that many decay profiles are not back to the baseline before the next pulse, in particularly S22b, S23b, S24b. I would suggest to re-do the measurement with lower repetition rate and add the IRF.

Author Reply: We thank the reviewer for the suggestion. We have now added the corresponding IRF (grey lines) and fitting curves (black lines) in revised Figures S18b, S19b, S20b, S22b, S23b, S24b.

Revision of Figure S18b, S19b, S20b in the revised supplementary information:

Revision of Figure S22b, S23b, S24b in the revised supplementary information:

On the other hand, we consider the time scale of Figs S18-S20b as sufficient as the signal is approximately ten times of the noise, which is the minimum considered appropriate for a good fit. On the other side, the reviewer is right for Figs. S22-24b. Here, however, we have used the temperature thin film data only in a qualitative manner without detailed numerical analysis. In any case, we re-measured the temperature-dependent lifetimes of both solution and crystalline films using an Edinburgh Instruments FLS980 spectrometer, as shown in the following. In fact, the data and trends of crystalline films are consistent with Figure S22b, S23b, S24b.

On the other side, for solution, the changes of short component are less pronounced as the detection scale becomes longer (as shown below).

Comment 9: Why did the paper not show any fitting curve of the decay profiles?

Author Reply: We thank the reviewer for the critical comment. For the main text, to avoid overcrowded plots, we decided to put only the main results of the fits in Table 1. On the other hand, exemplary fitting curves for selected temperatures are added in Figures S17, S18b, S19b, S20b, S22b, S23b, S24b (black lines).

Comment 10: There is clearly some raising component in S22b, S23b, S24b, as they are not showing sharp peaks at the excitation pulse (if you show the IRF). What is the raising component in the lifetime profiles? Is it the repopulation of LE?

Author Reply: As mentioned above, we have now re-done the temperature-dependent lifetime measurement of crystalline films using an Edinburgh Instruments FLS980 spectrometer. We note that the new measurements are essentially reproducing the former ones. The possible appearance of a rise time is considered as an outcome of the complex kinetics in the current system, which follows a consecutive-competitive reaction with a reversible step, as mentioned in the manuscript, and induce rise times for certain excitation and emission conditions. We desist however from further quantitative analysis, as the complete solution of the kinetics requires a higher number of observables than our experimental layout can provide.

Reviewers' Comments:

Reviewer #1:

Remarks to the Author:

We specifically pointed out two major points in the reviewing process of the original manuscript. The first point is related to the fluorescence decay analysis. But the authors admit that a glance at the experimental curves in polar solvents may be misleading; in fact, the percentages (fractional intensities) in Table 2 are a result of a deconvolution of the TCSPC data using the IRF. Comparison of the experimental curves with the IRF reveals that the curves at short lifetimes are dominated by the IRF. For this reason, the calculated fractional intensities look small. In this regard, the authors now add in the Table " Biexponential fits, obtained by deconvolution; fractional intensities are given in parentheses". Thus our inquiries have been well answered. The second point is the use of appropriate functionals to calculate the CT involved states in PBI systems. The authors have also demonstrated that the B3LYP functional indeed gives a much better reproduction of the spectral features in comparison with e.g.wB97XD or CAM-B3LYP. Thus, our inquiries have been addresses well in the revised manuscript. Thus, I do recommend the publication of this work in the current form.

Reviewer #2:

Remarks to the Author:

The authors have provided satisfactory responses regarding structural and photophysical studies. Thus, I would like to recommend the publication of this manuscript in its current form in Nature Communications.

Below are additional comments and suggestions:

- Please consider labeling the compound names directly on top or below the photographs in Figure 1b. This will be helpful for the reader to quickly understand the figure.
- It would be helpful to include additional figures in the SI to show how the donor's MOs are destabilized at different dihedral angles. I think the current Figure 4a/b is not clear enough to stress this point at the first glance.
- I appreciated the authors' attempt in trying to fabricate field-effect transistors devices, but unsuccessful to do so. I do hope to see the electrical properties of these PBI dyes to be reported in future studies.

Reviewer #3:

Remarks to the Author:

I was Reviewer 3. I thank the authors for their efforts in answering all my questions and comments. First of all, I need to clarify my first comment. I work with PBIs and understand how important they are in general. In my first comment, I asked what is the usefulness of the 4,6-ab-PBI, 6,8-ab-PBI and 8,10-ab-PBI and their unique properties (not other PBIs). The current draft fails to show the impact of these PBIs. By reading the paper, it seems the unusual optical property of the PBIs in this work is just a special case. I did not see any theory or methodology from this paper that I can adapt to other PBI derivatives or other chromophores. I can see the potential of this work, but the manuscript failed to deliver the usefulness of the special case, the logic of the molecular design and the impact of the methodology to the general audiences. I agree with the comment from Reviewer 2 that ' The current manuscript is more suitable for specialized journals.' I reject it for publishing in Nature Communications.

Please consider answering the following questions to improve the impact of this paper:
What is the logic behind the molecular design?

How can you control or optimize the theta in the future?

How can others use the special PBIs?

How do we adopt the methodology to other PBIs or other chromophores?

In addition, according to the paper (Figure 4), avoiding the CT state (from HOMO+1, +2 to LUMO) will increase the PLQY of the PBIs. However, what if one replaces the alkoxy chain structure or simply removes the N-benzyl? If the CT state is located on the benzyl alkoxy structure, why do you introduce the CT state to PBIs in your molecular design? It seems the paper designed a non-emissive pathway of the PBIs, and then tried to cancel it.

Reviewer # 1

General Comments:

We specifically pointed out two major points in the reviewing process of the original manuscript. The first point is related to the fluorescence decay analysis. But the authors admit that a glance at the experimental curves in polar solvents may be misleading; in fact, the percentages (fractional intensities) in Table 2 are a result of a deconvolution of the TCSPC data using the IRF. Comparison of the experimental curves with the IRF reveals that the curves at short lifetimes are dominated by the IRF. For this reason, the calculated fractional intensities look small. In this regard, the authors now add in the Table " Biexponential fits, obtained by deconvolution; fractional intensities are given in parentheses". Thus our inquiries have been well answered. The second point is the use of appropriate functionals to calculate the CT involved states in PBI systems. The authors have also demonstrated that the B3LYP functional indeed gives a much better reproduction of the spectral features in comparison with e.g. wB97XD or CAM-B3LYP. Thus, our inquiries have been addresses well in the revised manuscript. Thus, I do recommend the publication of this work in the current form.

We sincerely appreciate for the positive evaluation of our work. We would like to thanks for the reviewer's constructive comments, being all very helpful for improving the manuscript.

Reviewer # 2

General Comments:

The authors have provided satisfactory responses regarding structural and photophysical studies. Thus, I would like to recommend the publication of this manuscript in its current form in Nature Communications.

Author Reply: We are grateful for the reviewers' kind suggestions. We have revised the manuscript according to the reviewers' comments below.

Comment 1: Please consider labeling the compound names directly on top or below the photographs in Figure 1b. This will be helpful for the reader to quickly understand the figure.

Author Reply: We have redrawn this figure, and now labeled the compound names on top the photographs in Figure 1b.

Comment 2: It would be helpful to include additional figures in the SI to show how the donor's MOs are destabilized at different dihedral angles. I think the current Figure 4a/b is not clear enough to stress this point at the first glance.

Author Reply: To stress how the donor's MOs are destabilized at different dihedral angles, we have added detailed frontier MO energy levels for the different angles in the Supplementary information as follows.

Supplementary Fig. 28 | Frontier MO energy levels and topologies for the different angles.

Reviewer # 3

General Comments:

I was Reviewer 3. I thank the authors for their efforts in answering all my questions and comments.

First of all, I need to clarify my first comment. I work with PBIs and understand how important they are in general. In my first comment, I asked what is the usefulness of the 4,6-ab-PBI, 6,8-ab-PBI and 8,10-ab-PBI and their unique properties (not other PBIs).

The current draft fails to show the impact of these PBIs. By reading the paper, it seems the unusual optical property of the PBIs in this work is just a special case. I did not see any theory or methodology from this paper that I can adapt to other PBI derivatives or other chromophores.

I can see the potential of this work, but the manuscript failed to deliver the usefulness of the special case, the logic of the molecular design and the impact of the methodology to the general audiences. I agree with the comment from Reviewer 2 that 'The current manuscript is more suitable for specialized journals.' I reject it for publishing in Nature Communications.

Author Reply: We thank this reviewer for the detailed concerns about the manuscript. In any case, we would like to stress once more, that the majority of PBI black pigment

analogues, i.e. with dark red to black appearance usually are not luminescent, a behavior associated to strong π -stacking, which opens effective quenching pathways. This makes black pigment analogues by now only interesting for coloring purposes and charge transport, however not for OLEDs, OLETs, etc. For these reasons, our result on highly luminescent deep-red PBI pigments, despite strong π -stacking, appears as a very relevant finding.

Although we cannot demonstrate the device application of these molecules, we expect that such slice crystals with both high luminescence and high mobility would be an ideal starting material for organic light-emitting transistors (OLETs). Indeed, we tried to fabricate OFET devices based on the slice crystals but failed, and thus requires further optimization, both of the materials and of the device architecture. Such somewhat tedious device engineering issues however should not delay the publication of our first report on a considerable progress in materials design.

Comments from the reviewer:

Please consider answering the following questions to improve the impact of this paper:

What is the logic behind the molecular design?

How can you control or optimize the theta in the future?

How can others use the special PBIs?

How do we adopt the methodology to other PBIs or other chromophores?

In addition, according to the paper (Figure 4), avoiding the CT state (from HOMO+1, +2 to LUMO) will increase the PLQY of the PBIs. However, what if one replaces the alkoxy chain structure or simply removes the N-benzyl? If the CT state is located on the benzyl alkoxy structure, why do you introduce the CT state to PBIs in your molecular design? It seems the paper designed a non-emissive pathway of the PBIs, and then tried to cancel it.

Author Reply: We thank the reviewer for the critical adds, which we would like to answer together as a whole.

Starting point for our design concept was the surprisingly high 10% PL QY of the mb-PBI pigment, which however suffered from low solubility; thus, longer branched side chains seemed a logical route towards this goal. The high PL QY up to 60% for the new compounds, and, on the other side, the strong dependency on the side chain composition was indeed puzzling for us in the first place; only the successful x-ray analysis gave evidence for distinct structural differences, which were clearly correlated with the PL efficiency. Nevertheless, only the TD-DFT analysis enabled us to establish a model, which was able to consistently rationalize not only the different solid state behavior but also the (overall equally complex) behavior in solution, with its pronounced polarity dependence of the PL QY.

We note that the appearance of the CT state and the LE/CT interconversion evidently establishes an effective radiative pathway, which circumvents the quenching pathways commonly observed in PBI black pigment analogues. Therefore, simplification of the structures, as suggested by the reviewer, were – at least by now – not leading to highly luminescent deep-red PBI pigments. In this sense, despite the structural and mechanistic complexity, our approach appears, for the first time, as a viable route towards this demanded class of materials.

It may be important in the future not only to systematically vary the alkyl-side chains but to explore the influence of electronically directing co-substituents, as well as to modulate the molecular backbone through other rylene moieties.

Due to the unique properties of the current PBIs, there might be indeed further applications besides the obvious lighting applications in OLEDs and OLETs; for instance, solar concentrators, fluorescence imaging by NIR-emitting nanoparticles, or photovoltaics might be worth to explore.

All above information was now introduced in the manuscript as follows:

In the abstract, we now write "This effectively controls the emission process, and enables high Φ_F by circumventing the common quenching pathways commonly observed for perylene black analogues".

In the introduction, we added

"Therefore, the complex mb-PBI structure concept appeared very promising to generate emissive perylene black analogues, compared to earlier attempts with less complex structure modulation. "

and

"..., also offering better solubility in comparison with **mb-PBI**."

as well as

"Systematic photophysical studies in solution and solid state under varying environmental conditions, combined with computational studies fully rationalize the experimental results. Accordingly, we have discovered a long sought-after packing structure for this important class of pigments, now providing highly fluorescence solid state deep-red emitters."

In the Summary, we extended the text as follows: "Most importantly, this allows to create highly desired emissive deep-red solid-state emitters, despite close π -stacking, which commonly leads to quenching in perylene black analogues."